# Reprogramming Human Adult Fibroblasts into GABAergic Interneurons

**DOI:** 10.3390/cells10123450

**Published:** 2021-12-08

**Authors:** Andreas Bruzelius, Srisaiyini Kidnapillai, Janelle Drouin-Ouellet, Tom Stoker, Roger A. Barker, Daniella Rylander Ottosson

**Affiliations:** 1Group of Regenerative Neurophysiology, Lund Stem Cell Center, Department of Experimental Medical Science, Faculty of Medicine, Lund University, 221 84 Lund, Sweden; andreas.bruzelius@med.lu.se (A.B.); srisaiyini.kidnapillai@med.lu.se (S.K.); 2Faculty of Pharmacy, Université de Montréal, Montreal, QC H3T 1J4, Canada; janelle.drouin-ouellet@umontreal.ca; 3Wellcome-MRC Cambridge Stem Cell Institute and John van Geest Centre for Brain Repair, Department of Clinical Neurosciences, University of Cambridge, Cambridge CB2 0PY, UK; tbstoker@googlemail.com (T.S.); rab46@cam.ac.uk (R.A.B.)

**Keywords:** direct reprogramming, electrophysiology, iN, calbindin, adult skin cells, neuronal conversion, disease-modelling

## Abstract

Direct reprogramming is an appealing strategy to generate neurons from a somatic cell by forced expression of transcription factors. The generated neurons can be used for both cell replacement strategies and disease modelling. Using this technique, previous studies have shown that γ-aminobutyric acid (GABA) expressing interneurons can be generated from different cell sources, such as glia cells or fetal fibroblasts. Nevertheless, the generation of neurons from adult human fibroblasts, an easily accessible cell source to obtain patient-derived neurons, has proved to be challenging due to the intrinsic blockade of neuronal commitment. In this paper, we used an optimized protocol for adult skin fibroblast reprogramming based on RE1 Silencing Transcription Factor (REST) inhibition together with a combination of GABAergic fate determinants to convert human adult skin fibroblasts into GABAergic neurons. Our results show a successful conversion in 25 days with upregulation of neuronal gene and protein expression levels. Moreover, we identified specific gene combinations that converted fibroblasts into neurons of a GABAergic interneuronal fate. Despite the well-known difficulty in converting adult fibroblasts into functional neurons in vitro, we could detect functional maturation in the induced neurons. GABAergic interneurons have relevance for cognitive impairments and brain disorders, such as Alzheimer’s and Parkinson’s diseases, epilepsy, schizophrenia and autism spectrum disorders.

## 1. Introduction

Direct cell reprogramming offers unique access to human neurons from defined groups of patients for disease-modelling in vitro. With direct reprogramming, a somatic cell can be turned into a neuron using key combinations of transcription factors, miRNA and/or a cocktail of chemical compounds that manipulate pathways involved in neuronal fate and functions [1]. Moreover, direct reprogramming offers new approaches for disease modelling, as it is possible to reprogram skin cells from a human patient group into subtype-specific neurons and thereby generate human neurons with genetic backgrounds that are known to result in a particular disease. With such a human cell-based model, it is possible to also study idiopathic forms of disease [2]. Furthermore, given that these neurons retain the markers of aging, they are ideal candidates for modelling neuronal pathology in late-onset diseases, such as Parkinson’s and Alzheimer’s disease [3].

Thus far, several sources of non-neuronal cells have been converted into neurons. Glia cells, fibroblasts and pericytes have all been shown to be suitable for conversion into subtype-specific induced neurons (iNs) and have been generated from both mouse and human sources [4]. However, several factors, such as species and prolonged culturing of cells prior to conversion, limit the reprogramming efficiency. In particular, human cells are harder to reprogram than rodent cells [5,6], and cells from adult donors are much harder to reprogram than fetal cells [7,8]. This creates a barrier for using these cells for large-scale application and future clinical applications.

In this study, we have used a combination of fate-determining transcription factors together with RE1 Silencing Transcription Factor (REST) inhibition to remove the neuronal reprogramming barrier in adult dermal fibroblasts [9]. We have previously shown that REST inhibition is critical for a neuronal conversion of adult skin fibroblast [9], and therefore, REST inhibition was included in all reprogramming groups. Our combination of fate-determining transcription factors was aimed to derive subtype-specific iNs of a GABAergic phenotype; a population recently shown to be critically involved in neurological disorders of the aged brain, including Alzheimer’s [10] and Parkinson’s disease [11] but also epilepsy, autism spectrum disorders (ASD) and schizophrenia [12]. The transcription factor cocktail included five different genes involved in GABAergic fate development that have previously been shown to convert mouse and human fetal fibroblasts into GABAergic neurons [13], although have not yet been tested on human adult fibroblasts.

Our results showed that the human adult fibroblasts could be successfully converted into neurons using five different combinations of the transcription factors together with REST suppression. Furthermore, fibroblasts were also successfully converted into GABAergic neurons, as confirmed by the presence of GABAergic proteins and upregulation of GABAergic genes. Whereas a five-factor combination has previously been shown to convert mouse and human fetal fibroblasts [13], our study demonstrates that a two- to three-factor combination was superior in converting dermal fibroblasts from aged individuals into iNs with a GABAergic interneuron phenotype—more specifically, calretinin and calbindin interneurons.

This is the first demonstration of a human adult dermal fibroblast conversion into GABAergic interneuron and constitutes a step towards a more clinically applicable cell source for disease modelling of interneuron pathologies in the aging brain.

## 2. Materials and Methods

### 2.1. Fibroblast Culture

Dermal fibroblasts from adult individuals were obtained from the Parkinson’s Disease Research Clinic at the John van Geest Centre for Brain Repair (Cambridge, UK) and used under local ethical approval (REC 09/H0311/88). For more detailed information about biopsy sampling, see [9]. Thawed fibroblasts were plated and expanded in T75 flasks using standard fibroblast medium [Dulbecco’s modified Eagle’s medium (DMEM), 10% fetal bovine serum (FBS) and 1% penicillin–streptomycin] at 37 °C in 5% CO_2_. At confluency, cells were detached and dissociated using 0.05% trypsin, followed by plating at a lower density. In this study, fibroblasts from three healthy female donors were used, aged 53, 70 and 75 at the time of skin biopsy.

### 2.2. Viral Vectors

This study utilized a dual regulation system previously used in [13,14], consisting of a mix of inducible promoter guided- and constitutive promoter guided-transcription factors. Third-generation lentiviral vectors (LVs) expressing open reading frames (ORFs) for Ascl1 (mouse), Dlx5 (human) and Lhx6 (human) under the phosphoglycerate kinase (PGK) promoter were used for the constitutively expressed vectors, while Sox2 (mouse) and Foxg1 (human) were inducible under the doxycycline-regulated system (TET-ON promotor). Additionally, when dual regulation was utilized, a trans-activator (FUW.rtTA-SM2, Addgene, Watertown, MA, USA) was added. All transductions were accompanied by LVs expressing two separate constructs of shRNAs targeting the REST complex, under the control of a U6 promoter. Production of LVs was performed as previously described in [15] and titrated using a quantitative PCR analysis [15]. All titers were in the range of 3 × 10^8^ and 5 × 10^9^ transduction units/mL.

### 2.3. Neuronal Reprogramming

Prior to reprogramming, fibroblasts were plated at a density of 27,800 cells/cm^2^ in 48-well plates (Nunc, Roskilde, Denmark) using a 3-day coating process, starting with polyornithine (10 μg/mL, Sigma-Aldrich, St. Louis, MO, USA) overnight, laminin (5 μg/mL; Thermo Fisher, Waltham, MA, USA) overnight and fibronectin (0.5 ng/μL, Thermo Fisher, Waltham, MA, USA) overnight. For electrophysiological recordings, cells were plated directly onto serially coated glass coverslips pre-treated according to [16]. The conversion protocol was carried out as described previously [17]. In brief, cells were exposed to viral vectors the following day after seeding. Three days after transduction, the media was changed completely to neuronal differentiation media (NDiff227; Takara-Clontech, Kusatsu, Shiga, Japan) containing doxycycline (2 μg/mL, Duchefa, Haarlem, The Netherlands) and the small molecules CHIR99021 (2 μM, Axon, Groningen, Netherlands), SB-431542 (10 μM, Axon, Groningen, The Netherlands), noggin (0.5 μg/mL, R&D Systems, Minneapolis, MN, USA), LDN-193189 (0.5 μM, Axon, Groningen, The Netherlands) and valproic acid sodium salt (VPA; 1 mM, Merck Millipore, Burlington, MA, USA); and the growth factors LM-22A4 (2 μM, R&D Systems, Minneapolis, MN, USA), GDNF (2 ng/mL, R&D Systems, Minneapolis, MN, USA), NT3 (10 ng/mL, R&D Systems, Minneapolis, MN, USA) as well as db-cAMP (0.5 mM, Sigma-Aldrich, St. Louis, MO, USA). When conversion reached day 18, the small molecules were removed, and neuronal differentiation media was supplemented only with growth factors and db-cAMP. During the reprogramming period, two-thirds of the media was replenished every 2–3 days.

### 2.4. Real-Time Quantitative Reverse Transcription Polymerase Chain Reaction (qRT-PCR)

Total RNA was extracted using the RNeasy Mini Kit (Qiagen, Hilden, Germany), followed by reverse transcription from 200–300 ng total RNA from each sample using the Maxima First Strand cDNA Synthesis Kit (Thermo Fisher, Waltham, MA, USA) as per user instructions. Reverse transcription reactions were performed at 25 °C for 10 min followed by 50 °C for 15 min, and the reaction was terminated by heating to 85 °C for 5 min using a C1000 Touch Thermal Cycler (Bio-Rad, Hercules, CA, USA).

A Bravo Automated Liquid Handling Platform (Agilent, Santa Clara, CA, USA) was used to pipette real-time quantitative PCR (qPCR) reaction mixtures consisting of cDNA (1 μL), LightCycler^®^ 480 SYBR Green I Master mixture (5 μL, Roche, Basel, Switzerland) and relevant primers (4 μL, Eurofins, Ebersberg, Germany; See Appendix A) per well in a 384-well LightCycler^®^ 480 Multiwell plate (Roche). qPCR was performed using a LightCycler 480 II instrument (Roche) at 95 °C for 10 min, followed by 40 cycles at 95 °C for 30 s and 60 °C for 60 s in triplicate along with no template control (NTC). For each sample, the relative gene expressions were calculated using the ∆∆CT method against two housekeeping genes (ACTB and GAPDH), normalized to the expression of the genes in Day 0 fibroblasts and presented as a relative fold change in arbitrary units (AU).

### 2.5. Immunocytochemistry

Prior to immunocytochemistry, cells were fixed with 4% paraformaldehyde at room temperature (RT) for 10 min and rinsed three times with phosphate buffered saline (PBS); coverslips intended for GABAergic staining were fixed with 4% paraformaldehyde with the addition of 0.25% glutaraldehyde (cat. no. G6257, Sigma-Aldrich, St. Louis, MO, USA). Cells were further incubated in blocking solution containing 0.1% Triton X-100 (Sigma-Aldrich, St. Louis, MO, USA) and 5% donkey serum diluted in PBS for 1 h before the addition of primary antibodies (See Appendix A) in blocking solution; cells were then incubated at 4 °C overnight. After incubation with the primary antibody, cells were washed three times with PBS and incubated with secondary fluorophore-conjugated antibodies (1:200, Jackson ImmunoResearch Laboratories, West Grove, PA, USA) and DAPI (1:500, Sigma-Aldrich, St. Louis, MO, USA) diluted in blocking solution for 1 h at RT and, thereafter, washed three times with PBS prior to microscopy.

### 2.6. Microscopy

Fluorescent image acquisition was done using a Leica DM16000B widefield inverted microscope with Leica LAS X software (Leica, Wetzlar, Germany). Adobe Photoshop 2020 (Adobe, San Jose, CA, USA) was used for image processing, with adjustments applied equally over the entire image without loss of information.

### 2.7. High-Content Screening

Cells were analyzed using the Cellomics Array Scan (Array Scan VTI; Thermo Fisher Scientific, Waltham, MA, USA). Images were taken under a 10× magnification, and the total number of cells per well with nuclei and a single or double positive for either TUJ1, MAP2, GABA, calbindin and calretinin were quantified using the bioapplication “Target Activation” in HCS Studio 3.0 scan software (Thermo Fischer, Waltham, MA, USA). TUJ1+ neurons were selected based on a valid DAPI+ nuclei and a 488-TuJ1 average fluorescent intensity above that of an internal negative control. GABA+, calbindin+ and calretinin+ neurons were quantified as double positive 488-TuJ1 with either Cy3-GABA, Cy3-calbindin or Cy3-calretinin average fluorescent intensity above that of internal negative controls. Conversion efficiency was calculated as the percentage of TuJ1+/DAPI+ cells.

### 2.8. Electrophysiology

Electrophysiological recordings were performed at 25 days post-conversion. Glass coverslips with converted cells were transferred to a recording chamber and supplied with a constant flow of Krebs solution gassed with 95% O_2_ and 5% CO_2_ at RT. The composition of the Krebs solution was (in mM): 119 NaCl, 2.5 KCl, 1.3 MgSO_4_, 2.5 CaCl_2_, 25 glucose and 26 NaHCO_3_ with the pH adjusted to 7.4. For whole-cell patch-clamp recordings, a Multiclamp 700B amplifier (Molecular Devices, San Jose, CA, USA) was used together with borosilicate glass pipettes (3–7 MOhm) backfilled with the following intracellular solution (in mM): 122.5 potassium gluconate, 12.5 KCl, 0.2 EGTA, 10 Hepes, 2 MgATP, 0.3 Na_3_GTP, and 8 NaCl adjusted to pH 7.3 with KOH as in [14] pClamp 10.2 (Molecular Devices, San Jose, CA, USA), which was used for data acquisition with the current filtered at 0.1 kHz and digitized at 2kHz. The resting membrane potential was recordered in the current-clamp mode immediately after break in, followed by current injection to stabilize cells at a membrane potential of −60 to −70 mV. Current was injected from −20 pA to +35 pA with 5 pA increments for evoked action potentials. Inward sodium and delayed rectifying potassium currents were measured with the cells clamped at −70 mV in voltage-clamp using voltage-depolarizing steps supplied for 100 ms at 10 mV increments. Data were processed and analyzed with the Clampfit 10.3 (Molecular Devices, San Jose, CA, USA) and Igor Pro 8.04 (Wavemetrics, Portland, OA, USA) software combined with the NeuroMatic package [18].

### 2.9. Statistical Analysis

Statistical analysis was performed using a one-way analysis of variance (ANOVA), followed by Tukey’s multiple comparisons test or a Kruskal–Wallis test. All data are represented either as means or means ±SEM in which * *p* < 0.05; ** *p* < 0.01; *** *p* < 0.001; **** *p* < 0.0001. Statistical analyses were conducted using GraphPad Prism 9.0.0 (GraphPad, San Diego, CA, USA).

## 3. Results

### 3.1. Adult Fibroblast to Neuron Conversion Using Different Combination of Factors

First, we examined the individual contributions of fate-specifying transcription factors in the fibroblast-to-GABAergic interneuron conversion. In this screen, we tested five transcription factors involved in GABAergic development (Ascl1, Dlx5, Lhx6, Sox2 and Foxg1 (hereinafter referred to as A, D, L, S and F respectively). S and F were regulated by a dox-inducible promoter allowing expression for the first 14 days (Figure 1A), whereas A, D and L were regulated by constitutive promoters (Figure 1A) in order to mimic their temporal expression during embryonic development. All factors were either expressed individually or in different combinations, and each combination was tested on adult fibroblasts acquired from dermal biopsies of healthy human individuals aged 53–75. To further improve the overall conversion efficiency, the reprogramming cocktail was tested with short hairpin (sh) RNAs against the REST complex under a U6 promotor (Figure 1A), as this has previously been shown to remove the reprogramming barrier associated with adult cells [9]. During the timeline of the experiment, fibroblasts in all conditions gradually changed morphology towards a more rounded neuronal shape independent of factor combination (Figure 1B). At 25 days post-conversion, the morphology was drastically different from the starting population with a high number of cells now showing a rounded cell body and long extensions/processes indicative of a successful conversion. At this stage, we could confirm an effective REST inhibition in group A, AD and ADL, whereas the groups ADLF, ADLS and ADLSF showed no or minor downregulation (Appendix A). The upregulation of the individual transcription factors was seen in all relevant groups, although to a variable extent and with A, AD and ADL showing the highest upregulation of the transcription factor A (Appendix A).

In line with the change in morphology, gene expression analysis of fibroblasts transduced with different combinations of the five transcription factors displayed a decrease in fibroblast specific protein 1 (*FSP1*) at day 25 compared to non-converted cells, further supporting a change from the starting fibroblast phenotype. At 25 days post-conversion, the converted cells had an increased gene expression of microtubule-associated protein 2 (*MAP2*), synapsin (*Syn1*) and tubulin beta 3 class III (*TUBB3*)—three typical pan-neuronal genes, as compared to the parental fibroblasts (Figure 2C).

Immunocytochemistry further confirmed the expression of the neuronal protein beta-III-tubulin (TuJ1), as well as MAP2 in the iNs at 25 days post-conversion for all factor combinations (Figure 1D). The immunopositive iNs exhibited neuronal morphology and axonal projections (Figure 1D). Conversion efficiency analysis confirmed 14–50% of TuJ1+ cells across the different conditions (Figure 1E). Among the different factor combinations, A, AD and ADL showed the highest percentage of TuJ1+ cells (A; 50 ± 8.8% AD; 42 ± 8.3% and ADL; 39.6 ± 3.1) compared to other groups (Figure 1E). The lowest conversion rate was observed from combinations containing Sox2 (Figure 1E).

### 3.2. Human Adult Fibroblasts Convert into GABAergic iNs

The five transcription factors used for conversion, A, D, L, S and F, are highly expressed in the human medial ganglionic eminence (MGE), a major site of forebrain GABAergic neurons during embryogenesis [19], and the same factors have also been shown to be important for the differentiation and maturation of cortical interneurons [20]. We therefore sought to evaluate the GABAergic identity of the iNs.

First, the mRNA levels of GABAergic fate determinants were measured with qPCR at 25 days post-conversion. As expected, the mRNA levels of GAD2 were increased in all reprogramming conditions (Figure 2A), in line with the role of the transcription factors in GABAergic neuronal development. Moreover, the mRNA levels of NKX2.1, an MGE-enriched marker [21], and aristaless-related homeobox (ARX), a transcription factor expressed in forebrain interneurons regulating their migration and specification [13], were also increased in all groups. Although expressed at variable levels, this suggests that at least a subset of fibroblasts had converted into GABAergic neurons of the forebrain (Figure 2A). In addition, the mRNA levels of SLC17A7 (VGLUT1), a marker of glutamatergic neurons was increased, suggesting a possible glutamatergic population of iNs in the cultures.

In order to assess the GABA expression in the iNs, immunocytochemistry was performed at day 25 for all the different reprogramming conditions. Results showed that the iNs distinctly co-expressed the neurotransmitter GABA with TuJ1, supporting a highly enriched GABAergic phenotype in the iNs. When comparing the different combinations of transcription factors, A, AD and ADL, combinations showed greater co-expression with GABA, producing the highest efficiency, with as much as 74% of TuJ1 cells co-expressing GABA for the Ascl1-converted fibroblasts and 52% for the AD and ADL combinations. In contrast, we could not detect any protein expression of KGA glutaminase (a marker of glutamatergic neuron) in our iNs with immunocytochemistry in any of the groups.

Together, this indicates that the use of A, AD or ADL robustly converts human adult fibroblasts into a high proportion of GABA-producing neuronal cells. Given both the slightly higher neuronal conversion efficacy (Figure 1E) and the lack of REST inhibition combined with the higher proportion of GABA-producing iNs (Figure 2C), we decided to further evaluate only these three conditions for subtype identity and functional maturation.

### 3.3. Molecular Characterization of Human-Induced GABAergic Neurons

As the reprogramming genes used herein are specifically involved in the development of GABAergic forebrain interneurons, we next examined the protein expression for GABAergic interneurons using immunocytochemistry. Previous data from our lab has demonstrated the induction of calbindin+, and to some extent, parvalbumin+ interneurons from human glia using a similar combination of genes [14]. Therefore, we first assessed the protein expression of these markers. Results showed a significant expression of calbindin that was clearly co-localized with TuJ1 (Figure 3A). Quantification using cellomics demonstrated a slightly higher proportion of calbindin+ cells in the ADL condition (30.9 ± 10.5%) compared to both AD (21.9 ± 10.7%) and A condition (23.8 ± 5.7%) (Figure 3B). On the other hand, parvalbumin was rarely detected and not at sufficient levels for analysis.

Another major interneuronal subtype specifically prevalent in the human forebrain is the calretinin expressing neuron subtype [11]. This protein could also be detected in all three reprogramming groups A, AD and ADL, though to a lower extent compared to calbindin (Figure 3A,B). Quantification using cellomics demonstrated that the AD combination co-expressed calretinin and TuJ1 to a higher extent; 12.6 ± 7.6% compared to ADL; 2.7 ± 2.2%. For the A-condition, the levels of calretinin+ iNs were too low for quantification.

In order to assess additional molecular characteristics, we performed qPCR on the iNs transduced with A, AD or ADL for a spectrum of different interneuronal genes. Several of these genes were upregulated, including those of MGE-derived parvalbumin (*PV*) and somatostatin (*SST*) interneurons (Figure 3C). As expected, there was an upregulation of calbindin 1/calbindin (*CALB1*) and calbindin 2/calretinin (*CALB2*) in A-, AD- and ADL-converted iNs. Furthermore, there was a variable upregulation of a vasoactive intestinal polypeptide (*VIP*) and neuropeptide Y (*NPY*) could be seen in the AD and ADL groups [20], suggesting a wide range of interneuron characteristics in the iNs. In addition to interneuron genes, we could also detect some upregulation of the cAMP-regulated phosphoprotein (*DARPP32*) and t-box brain transcription factor 1 (*TBR1*), markers of striatal medium-spiny neurons and cortical projection neurons respectively.

In summary, our data suggests that adult human fibroblasts transduced with A, AD or ADL can be converted into GABAergic interneurons with the expression of relevant markers.

### 3.4. Conversion into Neurons with Functional Properties

To explore whether the iNs exhibited functional membrane properties similar to those of neurons, we performed patch-clamp electrophysiology recordings at day 25 when cells express mature and subtype-specific neuronal markers. Cells with a rounded cell body and neuronal characteristics were selected for whole-cell patch-clamp (Figure 4A). In voltage-clamp mode, the iNs showed neuron-specific, fast-inactivating inward and outward currents, which corresponds to the opening of voltage-gated sodium (Na+) and potassium (K+) channels, respectively (Figure 4B). Measurements of capacitance, while similar between groups, was variable between cells in each condition, suggesting a different maturation state of the iNs (Figure 4C). The resting membrane potential (RMP) remained more positive than for mature neurons and displayed no significant change between the conditions (Figure 4C). Together, these data show that iNs possess the fundamental structural components required to function as neurons, although at an immature state. Next, we elicited action potentials in current-clamp mode (Figure 4D). Among the cells patched, 2/7 (A), 8/13 (ADL) and 4/10 (AD) showed a current-induced action potential, revealing a successful transformation from a fibroblast into a neuronal phenotype. The action potentials were still immature with wider duration and lower amplitude, compared to those typically seen in mature neurons.

## 4. Discussion

Currently, there is an unmet need for cell-based models that capture human physiology in the context of aging. With the first report in 2010 [22], iNs emerged as a potential source of disease- and patient-specific neurons for disease modelling that retained the host age in the reprogrammed cells. Studies in the last decade have shown successful conversion of embryonic or early postnatal fibroblasts of mouse and human origin into iNs [6,8,9,23,24]. However, few studies have reported the generation of iNs from adult and aging cells [3,9,25]. iNs offer advantages in modelling adult-onset disorders, such as Parkinson’s-, Alzheimer’s- and Huntington’s disease [26] but also epilepsy and, to some extent, psychiatric disorders, as they retain age-associated markers of starting adult human fibroblasts, including the epigenetic age [27]. To better realize their full potential of these cell models, improved protocols are needed for the generation of subtype-specific and disease relevant neurons.

This study has focused on GABAergic interneurons that are implicated in epilepsy [28], Alzheimer’s disease [29] and Parkinson’s disease [30].

For this, we have compared a different combination of GABAergic fate determinants together with REST suppression to overcome the epigenetic barrier of reprogramming in aged human fibroblasts. Since REST can be seen as an adult-specific barrier for reprogramming of not only adult fibroblasts [9] but also astrocytes [31] and mouse or human fetal fibroblasts [5,32], we chose to include it in all our factor combinations. The downregulation of REST was efficient for the groups A, AD and ADL, whereas it was not as efficiently suppressed in groups ADLS, ADLF and ADLSF. This could be explained by the increased load of viral transcription factors in these conditions and further supports the necessity of REST inhibition in neuronal conversion of an adult fibroblast [9].

While most studies on patient fibroblast-based conversion have shown a pan-neuronal or unspecified neuronal subtypes [2], we have in this study evaluated subtype-specific markers in the iNs and found a relatively high GABAergic proportion with upregulation of a broad repertoire of interneuron-specific genes. In contrast and despite upregulation of vGluT1 mRNA in the iNs, we could not detect glutamatergic protein in the cultures, similar to a previous study [13]. This suggests a GABAergic-biased fate specification compared to the glutamatergic fate in the converted human neurons, at least at this early stage. In conclusion, subtype-specific reprogramming is an important step forward, as it is likely that iNs of different subtypes may express distinct disease-related phenotypes dependent on the brain area and neuronal population affected by the disease. Therefore, the ability to generate GABAergic neurons from adult human fibroblasts from patient skin biopsies provides a new and improved cell system for studying age-related disorders in people.

Our results showed the presence of some specific calcium-binding proteins associated with interneurons, such as calretinin and calbindin, in a comparatively high proportion of cells. Specifically, calretinin subtype outnumbers the other interneuronal subtypes found normally in the human and primate brain compared to mice [33], indicating a potentially important role for them in higher species. In support of this, the calretinin interneuron subtype has been linked to epilepsy pathophysiology, further supporting the clinical relevance of the reprogramming protocols described here [34].

One important caveat with most adult fibroblast conversion methods is that the iNs seem to be at an early stage of conversion. Thus, rather immature neurons (in terms of function) are typically generated, probably due with lower expression of voltage-gated sodium channels [2,35]. By including REST knock-down in our conversion method, we were able to show neuronal function with action potentials and inward/outward currents in a portion of iNs at 4 weeks, similar to a previous study looking at adult fibroblast conversion [6]. The functional maturity of human fibroblasts converted to neurons has previously been reported to develop over longer time points [35], and this may be enhanced when iNs have been co-cultured with astrocytes [7]. Moreover, to improve long-term physiological maturation of converted cells, alternative culture systems [36] or transplant models should be explored.

Our protocol differs slightly from previous published protocols [13,14] by not depending on Sox2 and Foxg1 for the generation of GABAergic interneurons. Instead, the conversion efficacy into GABAergic iNs seemed to improve with a reduced number of transcription factors and viral load in the cocktail. Whether or not these two early GABAergic fate determinants are necessary for the conversion seems to depend on the starting source of cells, i.e., adult fibroblast in this study vs. cells at a more developmental stage in a previous study, i.e., stem cell-derived glial cells or fetal fibroblasts.

In conclusion, we herein report that fibroblasts from aged individuals can successfully be converted into iNs by forced expression of GABAergic fate determinants together with REST knock-down. We identified three conversion factors, A, AD and ADL, that resulted in an efficient conversion into GABAergic neurons, with a conversion efficiency as high as 74%. By day 25, the iNs displayed mature neuronal morphologies, along with functional properties of neurons, such as the expression of voltage-gated ion channels and the ability to fire current-induced action potentials. As such, this study provides a new human model by which to study late-onset neurodegenerative and neuropsychiatric diseases that involve GABAergic interneurons [37]. Given the limited scope of current available therapies for these disorders, a better understanding of GABAergic interneuronal pathology could open up new innovative therapeutic opportunities [29].

## Figures and Tables

**Figure 1 cells-10-03450-f001:**
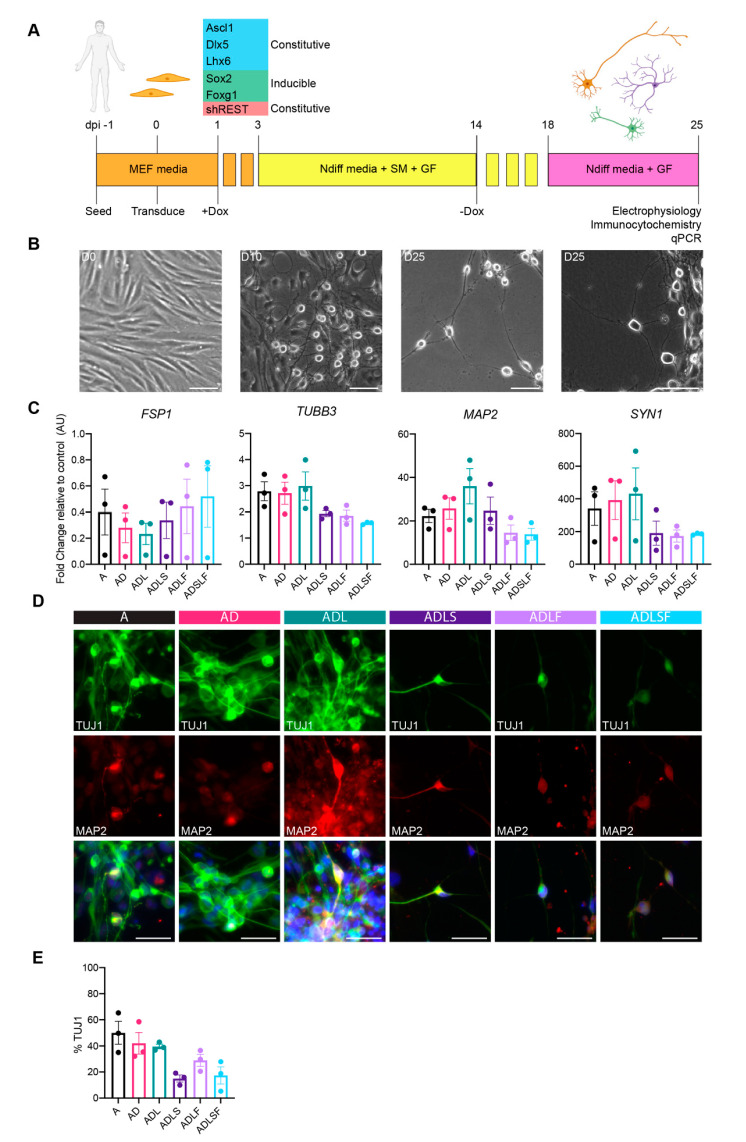
Conversion of adult human fibroblasts into neurons. (**A**) Schematic overview of the reprogramming protocol with different factor combinations. (**B**) Bright field images of cells in culture at day 0, day 10 and day 25. (**C**) RT-qPCR showing downregulation of fibroblast marker (FSP1) and upregulation of neuronal genes (TUBB3, MAP2 and SYN1) compared to fibroblast control across all conditions at day 25 (*n* = 3). (**D**) TuJ1 and MAP2 immunocytochemistry of reprogrammed neurons across all conditions at day 25. (**E**) Neuronal purity quantified as TuJ1+ cells across all conditions at day 25. Scale bars represent 50 µm.

**Figure 2 cells-10-03450-f002:**
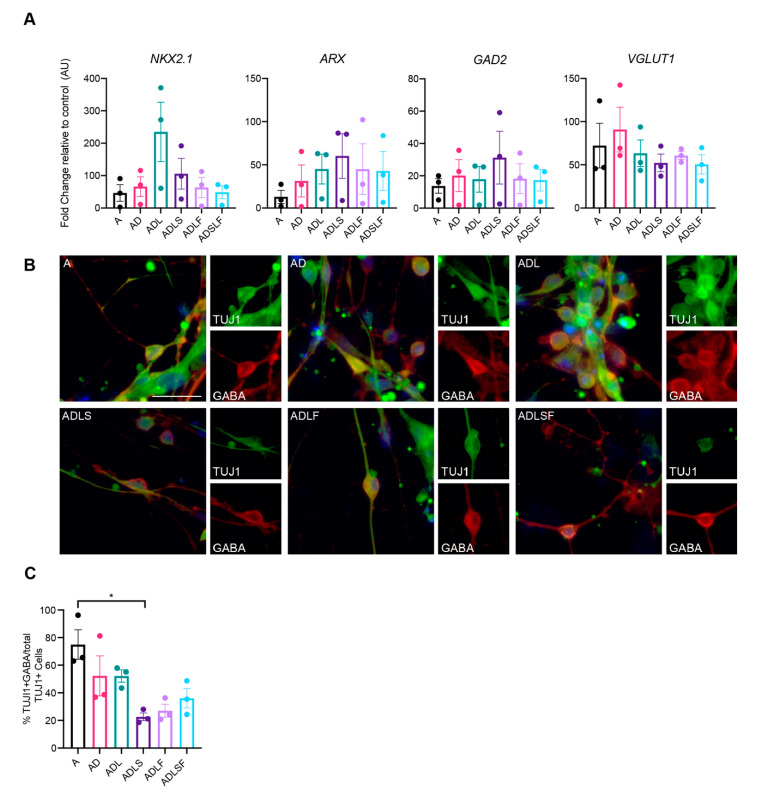
Quantification of GABAergic cells across different viral conditions at day 25. (**A**) RT-qPCR of neuronal lineage markers (*n* = 3). (**B**) Immunocytochemistry of reprogrammed GABAergic neurons. (**C**) Quantification of TuJ1 and GABA double positive cells (*n* = 3) * *p* < 0.05. Scale bar represents 50 µm.

**Figure 3 cells-10-03450-f003:**
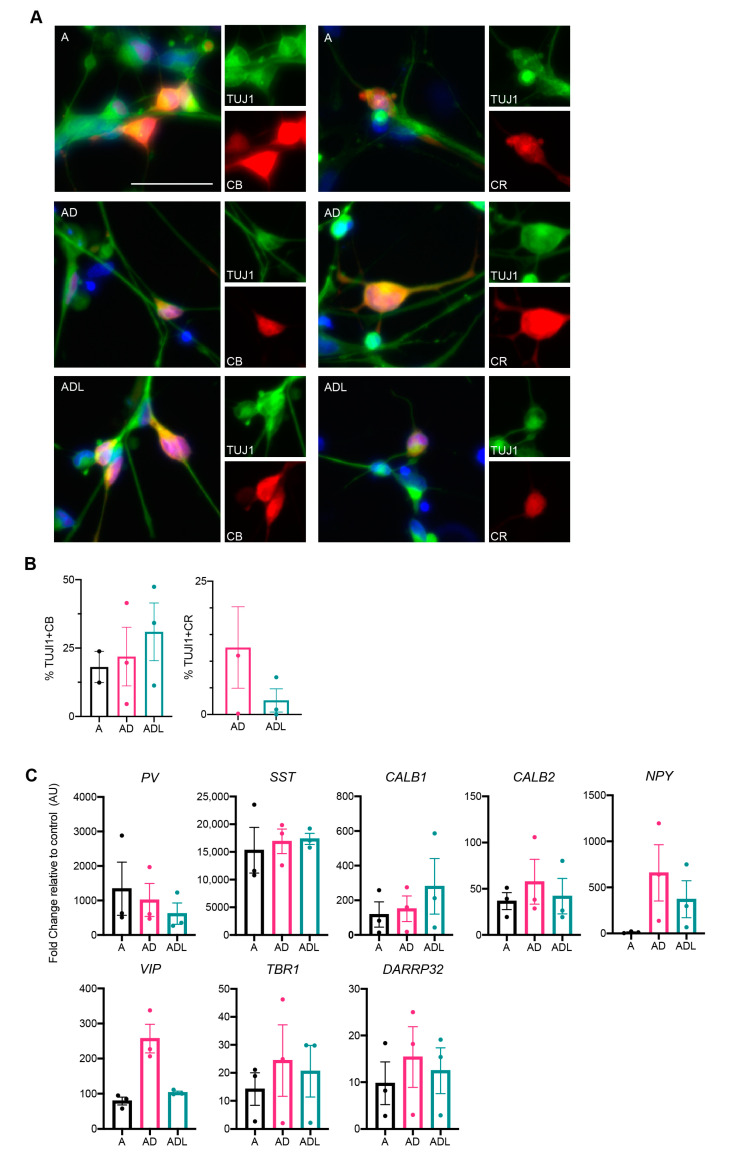
Characterization of the interneuron subtypes generated with A, AD and ADL conversion cocktail at day 25. (**A**) Immunocytochemistry of reprogrammed subtype-specific interneurons. (**B**) Quantification of TuJ1/CB (calbindin) and TUJ1/CR (calretinin) double positive cells at day 25 (*n* = 2 for A, *n* = 2–3 for AD and *n* = 2–3 for ADL). (**C**) RT-qPCR of subtype-specific lineage markers. Scale bar represents 50 µm.

**Figure 4 cells-10-03450-f004:**
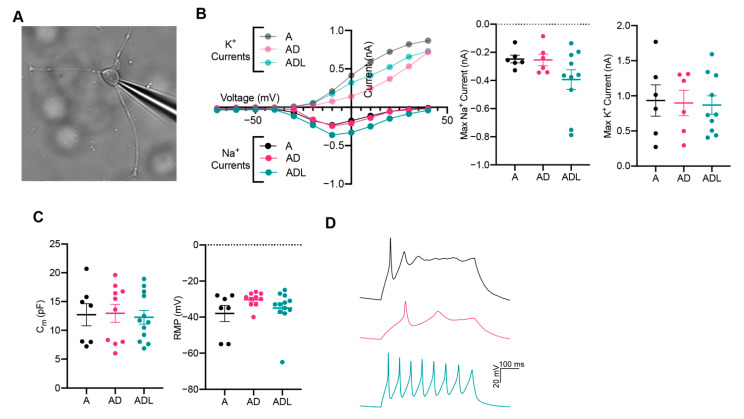
Functional analysis of induced neurons converted with A, AD or ADL at day 25. (**A**) Representative image of iN in whole-cell patch-clamp configuration. (**B**) Inward Na+ and outward K+ currents plotted against stepwise voltage induction at day 25: A (black), AD (magenta/pink) and ADL (green/light green); with each dot representing a mean value (*n* = 5 for A, *n* = 6 for AD and *n* = 10 for ADL). (**C**) Intrinsic membrane properties, capacitance (left) and resting membrane potential (RMP) (right). (**D**) Representative trace of immature evoked action potentials triggered by rheobase current injection steps: A (black), AD (magenta) and ADL (green).

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
