# Peer review of "Reprogramming Human Adult Fibroblasts into GABAergic Interneurons"

_cells, 2021, doi:10.3390/cells10123450_

Round 1

Reviewer 1 Report

The authors present a new method of converting adult human fibroblasts into GAGAergic interneurons with five transcription factors and REST inhibition. Based on immunofluoresecent staining, gene expression detection and electrophyisological analysis, the induced neurons were well characterized.

Author Response

We thank the reviewer for his/her time and for the positive feedback of our manuscript.

Reviewer 2 Report

In this study the authors set out to define reprogramming strategies that alleviate hurdles to cell fate transitions in adult skin fibroblasts. The key finding of this manuscript is that the knockdown of REST can alleviate the barrier to neuronal reprogramming. Overall the manuscript is well written and the data presented are convincing. However, there are some important missing controls. To demonstrate the central findings of this study more convincingly, the authors need to validate their constructs and provide more controls as follows:

  • Present a comparator set of control experiments without REST knockdown (shScrambled control) in Figure 1C and 1E.
  • Validate with at least one transcription factor (and/or combination) that the same increase in neurogenic conversion is obtained with a different shREST1/2 knockdown construct. This experiment would confirm that there are not off-target effects of their knockdown constructs. Using at least two shRNA constructs is an important experimental standard. I am not suggesting that all experiments be repeated, but at least one set.
  • Demonstrate that REST knockdown is efficient using qPCR and/or Western blot
  • Confirm that the individual transcription factors are effectively expressed by qPCR and/or Western blot

Specific points:

  1. lines 204-206. “…with short hairpin (sh) RNAs against the REST complex under a constitutive promotor (Figure 1A)…”

I am guessing the authors mean the U6 promoter for RNA Pol III transcripts? Instead of calling it a ‘constitutive” promoter, the authors should specify the identity of the promoter.

2. lines 212-214 “ In line with the change in morphology, gene expression analysis of fibroblasts transfected with different combinations of the five transcription factors displayed a decrease in fibroblast specific protein 1 (FSP1) compared to non-converted cells further  supporting a change from the starting fibroblast phenotype.”

Are these qPCR experiments at 25 days post conversion? Stage should be specified.

3. lines 215-217 “At 25 days post-conversion, the converted cells had a drastic increase in mRNA of Microtubule-Associated Protein 2 (MAP2), Synapsin (Syn1) and Tubulin Beta 3 Class III (TUBB3), three typical pan-neuronal genes, as compared to the parental fibroblasts (Figure 2C).”

The authors should change “a drastic increase” to stating fold-changes.

4. The axis in Figure 2 should be %TUJ1+GABA+/total TUJ1+ cells.

5. When describing Figure 2, the authors do not comment on the increase in VGLUT1, which is a marker of glutamatergic neurons shown in Figure 2A.

6. Are some neurons also glutamatergic? The authors should also quantitate glutamatergic neurons with immunostaining (similar to Figure 2C for GABA) by quantifying %TUJ1+vGLUT+/total TUJ1 cells.

Author Response

In this study the authors set out to define reprogramming strategies that alleviate hurdles to cell fate transitions in adult skin fibroblasts. The key finding of this manuscript is that the knockdown of REST can alleviate the barrier to neuronal reprogramming. Overall the manuscript is well written and the data presented are convincing. However, there are some important missing controls. To demonstrate the central findings of this study more convincingly, the authors need to validate their constructs and provide more controls as follows:

  • Present a comparator set of control experiments without REST knockdown (shScrambled control) in Figure 1C and 1E.

We thank the reviewer for his or her time and valuable input to our manuscript. We agree that a comparator set of control experiments without REST is a valid request. We have previously made these experiment in the lab using the very same adult human fibroblast lines, (see Drouin-Ouellet et al, EMBO, 2017) and demonstrated that REST knockdown is necessary for a neuronal conversion (see Figure 1H in Drouin-Ouellet with only reprogramming factors (Ascl1 + Brn2) and in Fig 2B with reprogramming factors + REST KD). Based on the this we are certain that our REST knockdown is needed to convert adult human dermal fibroblasts. 

We have now clarified these previous experiments in the text, see page 2, line 54.

  • Validate with at least one transcription factor (and/or combination) that the same increase in neurogenic conversion is obtained with a different shREST1/2 knockdown construct. This experiment would confirm that there are not off-target effects of their knockdown constructs. Using at least two shRNA constructs is an important experimental standard. I am not suggesting that all experiments be repeated, but at least one set.

We fully agree that is it important to confirm the conversion with another REST knockdown construct and apologize for the not clear information on this. We have in fact used two different constructs in our conversions with two different shRNAs targeting REST as in previous publications (Drouin-Ouellet et al, EMBO, 2017).

Furthermore, we have published another study showing how these specific constructs of shRNAs impacts the neuronal gene regulatory network (see Merlevede et al, Sci Rep, 2021). This study shows that REST KD is part of the same network as PTB and that you can achieve the same effect with a sh-nPTB KD. Based on this we trust that REST KD is specific.

We have now clarified this in materials and methods, page 3, line 101.

In addition, we have explored

  • Demonstrate that REST knockdown is efficient using qPCR and/or Western blot

We thank the reviewer for this valuable and important input. We have now performed qPCR of REST1/2 knockdown in the Fig. 1 Supplement and refer to this on page 5, line 218. Interestingly we were not able to knock-down the REST complex in the groups ADLS, ADLF and ADLSF which might be the explanation to why these groups did not convert as efficiently to neurons.

We comment on this on line 218, 279 and in the discussion 378.

Confirm that the individual transcription factors are effectively expressed by qPCR and/or Western blot.

We have confirmed an effectively expression by qPCR for Ascl1, Dlx2, Lhx6, Sox2, Foxg1 and added this to Supplement 1. In the manuscript see page 5, line 221.

Specific points:

  1. lines 204-206. “…with short hairpin (sh) RNAs against the REST complex under a constitutive promotor (Figure 1A)…”

I am guessing the authors mean the U6 promoter for RNA Pol III transcripts? Instead of calling it a ‘constitutive” promoter, the authors should specify the identity of the promoter.

We have change this, accordingly, see line 206.

  1. lines 212-214 “ In line with the change in morphology, gene expression analysis of fibroblasts transfected with different combinations of the five transcription factors displayed a decrease in fibroblast specific protein 1 (FSP1) compared to non-converted cells further  supporting a change from the starting fibroblast phenotype.”

Are these qPCR experiments at 25 days post conversion? Stage should be specified.

This has been added now on line 216.

  1. lines 215-217 “At 25 days post-conversion, the converted cells had a drastic increase in mRNA of Microtubule-Associated Protein 2 (MAP2), Synapsin (Syn1) and Tubulin Beta 3 Class III (TUBB3), three typical pan-neuronal genes, as compared to the parental fibroblasts (Figure 2C).”

The authors should change “a drastic increase” to stating fold-changes.

We have change this to “increased gene expression of”, see track changes.

  1. The axis in Figure 2 should be %TUJ1+GABA+/total TUJ1+ cells.

We have changed this accordingly.

  1. When describing Figure 2, the authors do not comment on the increase in VGLUT1, which is a marker of glutamatergic neurons shown in Figure 2A.

We apologize for this mistake. We have now added a comment on this on page 7, line 264.

  1. Are some neurons also glutamatergic? The authors should also quantitate glutamatergic neurons with immunostaining (similar to Figure 2C for GABA) by quantifying %TUJ1+vGLUT+/total TUJ1 cells.

We agree with the reviewer that this is a valid request. We have done immunocytochemistry using two different markers for glutamate neurons, i.e. vGluT1 and KGA-glutaminase and could not detect any of the proteins in our iNs in any of the groups. See attachment for KGA staining.

We have clarified this with a sentence on page 7, level 274.

Reviewer 3 Report

This article by Daniella Rylander Ottosson and collaborators describes an optimized protocol for the reprogramming of adult human skin fibroblasts into GABAergic neurons. Although the protocol is not completely original, as it relies on the inhibition of REST along with a combination of GABAergic fate determinants previously described to convert mouse and human embryonic fibroblasts to GABAergic interneurons, this is the first demonstration of the conversion of adult humans dermal fibroblasts in GABAergic interneurons phenotype.

This work is well organized, experiments are rigorously conducted and results support the hypothesis formulated by the authors. The weak part of the work concerns the poor originality and the lack of characterization of differentiated cells to highlight any phenotypes, other than interneurons.

To reinforce this part of the article, I suggest investigating the presence of other neuronal cell subtypes. 

Author Response

This article by Daniella Rylander Ottosson and collaborators describes an optimized protocol for the reprogramming of adult human skin fibroblasts into GABAergic neurons. Although the protocol is not completely original, as it relies on the inhibition of REST along with a combination of GABAergic fate determinants previously described to convert mouse and human embryonic fibroblasts to GABAergic interneurons, this is the first demonstration of the conversion of adult humans dermal fibroblasts in GABAergic interneurons phenotype.

This work is well organized, experiments are rigorously conducted and results support the hypothesis formulated by the authors. The weak part of the work concerns the poor originality and the lack of characterization of differentiated cells to highlight any phenotypes, other than interneurons.

To reinforce this part of the article, I suggest investigating the presence of other neuronal cell subtypes. 

We thank the reviewer for the positive judgement and the constructive feedback on our manuscript. We agree that to highlight other phenotypes in the induced neurons would increase the value of the study. On this request we have performed immunocytochemistry on other neuronal subtypes and specifically, glutamatergic (KGA) and medium-spiny neuron (DARPP32) where we have seen upregulation in the gene expression (Fig 2 in the manuscript). (We have used antibodies that have been validated and frequently used in our lab.) Nevertheless, none of these markers were present in our cultures (see attachment). This suggest that our converted neurons indeed show a GABAergic interneuron fate at least at this an early stage and when using GABAergic fate determinant in the reprogramming cocktail.

We have now added a sentence on this in the results line 274 and in the discussion, line 386.

Reviewer 4 Report

In this original research manuscript, Bruzelius et al. describe the direct reprogramming of human adult fibroblasts into GABA-interneurons by transgene expression of transcription factors together with REST-knockdown and supplementation with small molecules and growth factors. One important finding of the study was that only 3 factors together with REST-knockdown was necessary to convert the fibroblasts into GABA-interneurons.

The report is well-written and easy to follow. The English language is at appropriate for scientific journal level, only a minor spell check might be required -  I noticed only one spelling error (in Figure 1A: “constituative” instead of “constitutive”), which the authors should correct.

No major concerns with the manuscript were noticed by this reviewer; therefore, I would recommend the manuscript for publication.

Author Response

We thank the reviewer for this positive feedback and have corrected the spelling error.

Round 2

Reviewer 2 Report

The authors have adequately addressed my concerns. 

Reviewer 3 Report

The authors have adequately addressed my concerns. In my opinion this paper can now be accepted in this form